# ON THE ENTROPY CALIBRATION OF LANGUAGE MODELS

## ABSTRACT

Language models are trained with teacher forcing but are used autoregressively, so errors accumulate as more tokens are generated. This issue is well-studied but remains a fundamental problem that harms generation quality. Building on past work, we take the perspective that error accumulation is reflected in the model's entropy, so we can better understand and address it through the lens of *entropy calibration*. A language model is *entropy calibrated* if its entropy over generations, i.e. its confidence, matches the log loss it incurs on actual text. First, we find that models are indeed miscalibrated in practice: for base models across a range of sizes, entropy per step increases as more tokens are generated, leading to generations becoming incoherent over time. On the other hand, after instruction tuning, the largest models now have too little entropy (i.e. are overconfident), leading to a lack of diversity in model outputs. From a theoretical perspective, entropy calibration is difficult to attain because it is a global property of the entire generation process, which has an exponentially large output space. Per-step adjustments are tractable but fail to preserve the model's log loss, while global adjustments preserve log loss but are intractable. Our main theoretical contribution is to propose future entropy scaling, an adjustment to the next token probabilities that uses information about the future entropy of each token, i.e. the average entropy of continuations from that token. With additional assumptions, we prove that this adjustment calibrates the model while preserving log loss. While future entropy estimation is expensive, this result suggests that calibration and stabilization of the entropy should be possible without trading off model quality.

## 1 INTRODUCTION

Modern language models are trained with teacher forcing and achieve very low log loss when predicting one word at a time. However, when deployed, they are primarily used autoregressively, and low log loss does not guarantee strong autoregressive performance because errors accumulate over time as the model conditions on its own outputs. As a result, practitioners use various sampling tricks (e.g. temperature reduction, distribution truncation) to stabilize generation (Holtzman et al., 2020; Welleck et al., 2024). These tricks are applied ad hoc and it is not always clear when or why they are necessary.

In this paper, we build on the work of Braverman et al. (2020) and provide theory and experiments to better understand language model sampling through the lens of calibration. We say that a language model is *entropy calibrated* if its entropy over generations, i.e. its confidence, matches the log loss it incurs on actual text in expectation:

$$\mathbb{E}_{X \sim q}[\mathbb{E}_{Y \sim p^*(Y|X)}[-\log \hat{p}(Y \mid X)]] = \mathbb{E}_{X \sim q}[H_{\hat{p}}(\hat{Y} \mid X)], \tag{1}$$

where $q$ denotes the prompt distribution, $p^*$ is the true conditional distribution, $\hat{p}$ is the model, $X$ is the prompt, $Y$ is the response, and $H_{\hat{p}}(\hat{Y} \mid X)$ is the entropy of $\hat{p}$'s generation $\hat{Y}$ given the prompt $X$. If $\hat{p}$ has at most $\varepsilon$ KL divergence with $p^*$, calibration can also be thought of as requiring that the entropy of model generations be within $\varepsilon$ of the entropy of human text. The main premise of this paper is that many errors and instabilities in autoregressive generation are reflected in the model's entropy deviating from that of human text. Accordingly, sampling methods are effective if they correct miscalibration while preserving model quality. Using this framework, we find the following:

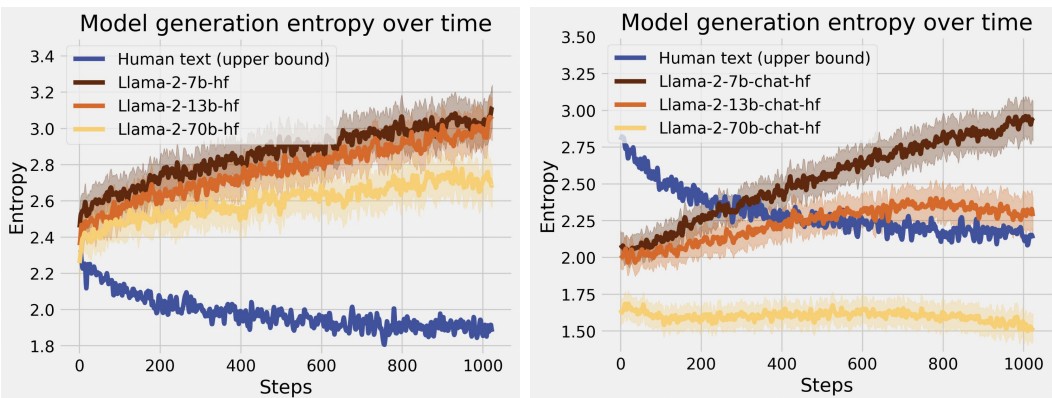

Figure 1: Left: entropy per step of base models; right: entropy per step of instruction-tuned models. The best model's log loss, which serves as an upper bound for the entropy of human text, is plotted in blue. In both plots, models are prompted with 128 tokens of context from a story from the `writingprompts` dataset and asked to generate 1024 additional tokens. Key takeaways: (1) For all base models, entropy per step increases over time, with stronger models starting lower but increasing at a similar rate. (2) After instruction tuning, smaller models still have too much entropy, while larger models now have too little entropy.

**We analyze current language models, finding that they are miscalibrated across model sizes:** In expectation, the entropy rate (i.e. entropy per time step) of human text is constant or decreases slightly over the length of a document (Genzel & Charniak, 2002; Verma et al., 2023). In contrast,

(a) For base models, entropy rate increases as more tokens are generated. As a result, outputs become incoherent over time. This result holds across model sizes: compared to weaker models, stronger models start at a lower entropy but still deviate upward at a similar rate (Figure 1).

   The fact that models become incoherent over time has been observed in past work (Holtzman et al., 2020), and practitioners use various truncation techniques to address this issue. We analyze the effect of these techniques on calibration and find that decreasing the sampling temperature shifts the entropy curve downward while also decreasing the slope; other truncation methods have a similar effect (Figure 4). However, as has been observed in prior work (Hashimoto et al., 2019; Zhang et al., 2021; Pillutla et al., 2021), this stabilization comes at the cost of model degradation in the form of increased log loss and reduced diversity (Figure 5).

(b) After instruction tuning, smaller models still have too much entropy, but larger models now have too little entropy (Figure 1). Miscalibration in the form of entropy being too low results in generations lacking in diversity and sometimes becoming repetitive over time. Furthermore, even for models whose entropy seems stable on average, individual generations still sometimes derail and are just counterbalanced by low-entropy generations (Figure 2).

   Existing methods are designed to decrease entropy, so they are not well-suited for calibrating large instruction-tuned models (Figure 6).

**We propose future entropy scaling and prove that it calibrates while improving log loss, suggesting that calibration is possible without model degradation:** Entropy calibration is difficult to achieve because it is a global property of the entire generation process: adjusting each generation step separately (e.g. with per-step temperature scaling) is tractable but harms log loss, while adjusting the entire generation process as a whole (e.g. with global temperature scaling) preserves log loss but is intractable because the output space is exponential (Braverman et al., 2020).

We prove that with additional assumptions, we can *tractably* calibrate entropy *while preserving log loss* by adjusting each token's probability based on what its future entropy would be. In particular, for a parameter $\alpha \in \mathbb{R}^T$ (where $T$ is the max generation length), let the *future-entropy-adjusted* model $\hat{p}_\alpha^{\text{ent}}$ be given by

$$\hat{p}_\alpha^{\text{ent}}(y_t \mid x, y_{<t}) = \text{softmax}\{(1 + \alpha_t) \log \hat{p}(y_t \mid x, y_{<t}) - \alpha_t H_{\hat{p}_\alpha^{\text{ent}}}(\hat{Y}_{>t} \mid x, y_{<t}, y_t)\}, \quad (2)$$

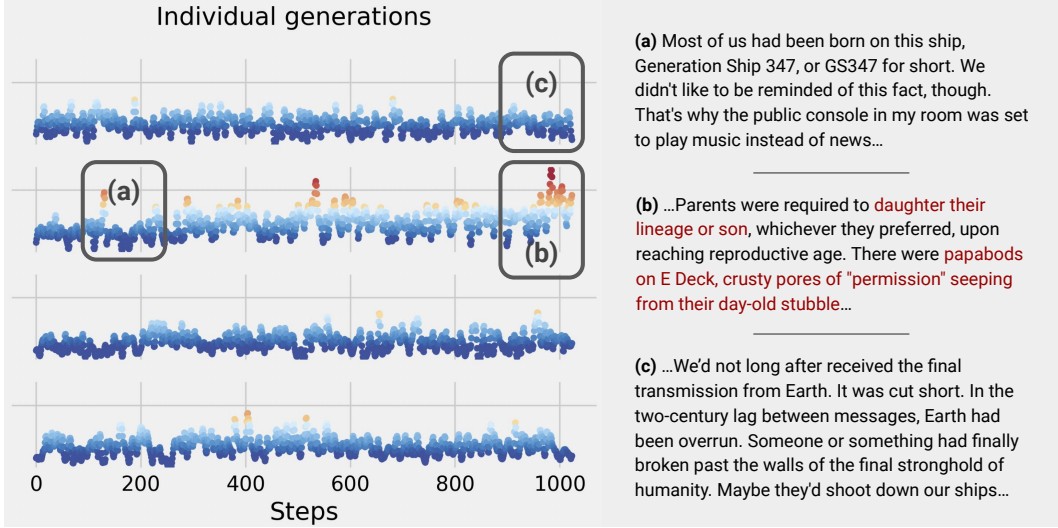

Figure 2: Four generations from `Llama-2-70B-chat-hf` for the same prompt, with plots of their entropy rate over time (blue: low entropy, red: high entropy). The model is prompted with the following instruction (along with 128 tokens from a human-written story): "Write a long story based on the following prompt: You are a part of the middle generation on a colony ship. You never saw Earth and will not see your destination." While the model has stable entropy rate on average (Figure 1), individual generations can still sometimes derail: the second sample is initially high quality (excerpt (a)) but has unstable entropy, leading to incoherent text (excerpt (b)). In contrast, the first sample's entropy remains stable, so it remains coherent until the end (excerpt (c)).

where future entropy $H_{\hat{p}_\alpha^{\text{ent}}}(\hat{Y}_{>t} \mid x, y_{<t}, y_t)$ denotes the total entropy of the entire continuation $\hat{Y}_{>t}$ if token $y_t$ were to be chosen. Intuitively, per-step adjustments which only look at next word probabilities are myopic, as any token that is generated also affects the remaining generation process. Therefore, to properly calibrate, one needs to anticipate how each token affects the future entropy.

We prove that choosing $\alpha$ to minimize log loss results in an adjusted model $\hat{p}_\alpha^{\text{ent}}$ that is entropy calibrated while having log loss at most that of $\hat{p}$. However, the future entropy of $\hat{p}_\alpha^{\text{ent}}$ is not tractable to compute in general. Therefore, the main assumption we need to make is that we can replace $H_{\hat{p}_\alpha^{\text{ent}}}$ with $H_{\hat{p}_\alpha'}$ for some surrogate model $\hat{p}_\alpha'$ whose future entropy behaves similarly to that of $\hat{p}_\alpha^{\text{ent}}$. In practice, we use $\hat{p}$ as the surrogate model, in which case we can estimate future entropy by averaging over samples. We describe this algorithm and its proof sketch in Section 5.1.

While estimating future entropy via sampling is expensive, this result suggests that (1) calibration is possible without trading off log loss, and (2) the main missing component in current methods is information about the entropy of future trajectories. By computing future entropy on a small set of examples, we also uncover interesting new failure cases of truncation-based samplers: while it is well known that truncation results in loss of diversity by suppressing perfectly good tokens, we also find cases where it fails to suppress tokens that, despite having moderate probability mass, can lead to degeneration (Figure 3). We discuss these examples along with other analyses in Section 5.2 and suggest potential opportunities for improving language model sampling.

## 2 RELATED WORK

**Entropy in models and text.** This paper draws upon a series of past works that study entropy in text generation. Genzel & Charniak (2002) use n-gram models to validate the *entropy rate constancy* principle, which posits that the entropy rate of human text is constant over time. Verma et al. (2023) revisit this hypothesis using neural language models and find more varied entropy patterns, but still find that after the first thirty or so tokens of a document, entropy rate is either constant or decreases slightly. Braverman et al. (2020) study the entropy of autoregressive model generations,

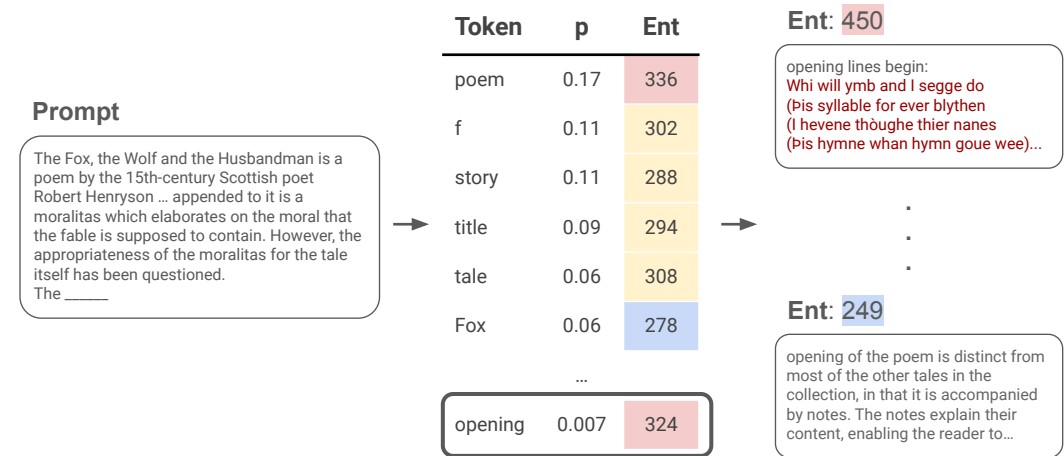

Figure 3: In this example from `TinyLlama_v1.1` applied to `wikitext-103`, each candidate next token is labeled with its probability under the base model, along with an estimate of its future entropy for 128 tokens (left: prompt, middle: candidate tokens, right: model generations). The highlighted token, "opening," has moderate probability and is not suppressed when sampling at temperature 0.9 (probability changes from 0.0070 to 0.0056). While the token is a reasonable one, it raises the difficulty of the subsequent generation because the model is tasked with generating a poem in Middle Scots, causing it to derail in roughly half of its continuations. In contrast, the correct adjustment, which takes future entropy into account, properly suppresses this token, reducing its probability from 0.0070 to 0.00013.

introducing the concept of entropy rate calibration. They first show that the entropy rate of language models increases over time, when it should ideally be time-invariant. Next, recognizing the global temperature scaling corrects miscalibration but is intractable, they instead propose a one-step lookahead algorithm that reduces miscalibration but only attains a one-step guarantee. We build on their work by proposing future entropy scaling, an algorithm that provably attains global entropy calibration. We also use entropy calibration to analyze current models and techniques, including base and instruction-tuned Llama models (Touvron et al., 2023) and various truncation-based samplers (Fan et al., 2018; Holtzman et al., 2020; Hewitt et al., 2022).

**Error accumulation in autoregressive generation.** The idea that autoregressive models accumulate errors during generation is well-known. Williams & Zipser (1989) introduce the term "teacher forcing" to refer to the technique of training neural models on only one generation step at a time, in contrast to autoregressive generation where the model must generate multiple steps in succession. To address this mismatch, also known as "exposure bias," a variety of papers propose alternate sequence-level training objectives (Ranzato et al., 2016; Welleck et al., 2020; Deng et al., 2020), but teacher forcing remains the dominant training method.

**Distribution truncation.** To stabilize autoregressive generation, a large number of truncation-based methods have been developed as alternatives to temperature scaling, including top-k sampling (Fan et al., 2018), nucleus (top-p) sampling (Holtzman et al., 2020), epsilon/eta sampling (Hewitt et al., 2022), and typical sampling (Meister et al., 2023). However, increased quality from truncation comes at the cost of diversity, and Hashimoto et al. (2019), Zhang et al. (2021), and Pillutla et al. (2021) propose methods to evaluate how well these methods perform this tradeoff. Basu et al. (2021) analyze how truncation parameters affect the entropy of the resulting sample, and use these insights to propose a method which dynamically sets these parameters during generation. Finally, Freitag et al. (2023), Shi et al. (2024), and Welleck et al. (2024) survey and compare sampling techniques across different models, datasets, and tasks, finding that the relative ranking between them is highly dependent on the setting.

**Calibration.** Model calibration is most commonly studied in binary classification, with some classic algorithms including binning, Platt scaling, and isotonic regression (Platt, 1999; Zadrozny & Elkan, 2002; Guo et al., 2017; Kumar et al., 2019). Entropy calibration can be thought of as a relaxation of multiclass calibration, where each class corresponds to a possible output string and the number of classes is exponential in the output length. Relaxing multiclass calibration to calibration of a loss function is related to the work of Zhao et al. (2021), who use a similar idea to define a calibration notion for multiclass classifiers in decision theoretic settings. In contrast with our setting, they consider settings like image classification where the number of classes is not exponential.

## 3 PRELIMINARIES

In this section, we define and provide intuition for entropy calibration, which was first proposed in Braverman et al. (2020). For notation, let $V$ denote the vocabulary, and let the prompt $X \in V^*$ and response $Y \in V^*$ be random variables taking values in $V^*$, the space of all strings over $V$. Also, let $X \sim q$ and $Y \sim p^*(Y \mid X)$ denote the ground truth prompt and response distributions, and let $\hat{p} : V^* \to \Delta^{|V|}$ be a language model mapping any string to a next token distribution over $V$. We will use $\hat{Y} \sim \hat{p}(\hat{Y} \mid X)$ to denote the response distribution induced by sampling autoregressively starting from the prompt $X$.

For a fixed prompt $X$, let $\mathcal{L}(p^* \parallel \hat{p}; X)$ denote the model's expected log loss on that prompt,

$$\mathcal{L}(p^* \parallel \hat{p}; X) = \mathbb{E}_{Y \sim p^*(Y|X)}[-\log \hat{p}(Y \mid X)]$$

$$= \mathbb{E}_{Y \sim p^*(Y|X)} \left[ \sum_{t=1}^{\text{len}(Y)} -\log \hat{p}(Y_t \mid X, Y_{<t}) \right], \tag{3}$$

and let $H_{\hat{p}}(\hat{Y} \mid X)$ denote the entropy of model generations on that prompt:

$$H_{\hat{p}}(\hat{Y} \mid X) = \mathbb{E}_{\hat{Y} \sim \hat{p}(\hat{Y}|X)}[-\log \hat{p}(\hat{Y} \mid X)]$$

$$= \mathbb{E}_{\hat{Y} \sim \hat{p}(\hat{Y}|X)} \left[ \sum_{t=1}^{\text{len}(\hat{Y})} -\log \hat{p}(\hat{Y}_t \mid X, \hat{Y}_{<t}) \right]. \tag{4}$$

Then, we say that $\hat{p}$ is *entropy-calibrated* if its entropy over generations, i.e. its confidence, matches the log loss it incurs on actual text in expectation:

$$\mathbb{E}_{X \sim q}[\mathcal{L}(p^* \parallel \hat{p}; X)] = \mathbb{E}_{X \sim q}[H_{\hat{p}}(\hat{Y} \mid X)]. \tag{5}$$

*Entropy calibration error* is then given by the difference between entropy and log loss, or

$$\text{EntCE}(p^* \parallel \hat{p}) = \mathbb{E}_{X \sim q}[\mathcal{L}(p^* \parallel \hat{p}; X) - H_{\hat{p}}(\hat{Y} \mid X)]. \tag{6}$$

The goal of calibration is to ensure that $\frac{1}{T}|\text{EntCE}(p^* \parallel \hat{p})| \leq \varepsilon$ after $T$ autoregressive generation steps, for some per-step miscalibration tolerance $\varepsilon$. A few notes about this definition:

(a) The model's log loss is an upper bound for the entropy of $p^*$, with bound being tighter if its KL divergence (i.e. excess log loss) is small: for KL divergence given by

$$\mathbb{E}_{X \sim q}[D_{KL}(p^* \parallel \hat{p}; X)] = \mathbb{E}_{X \sim q}[\mathcal{L}(p^* \parallel \hat{p}; X) - H_{p^*}(Y \mid X)], \tag{7}$$

we have that the KL is bounded by $0 \leq \mathbb{E}_{X \sim q}[D_{KL}(p^* \parallel \hat{p}; X)] \leq \varepsilon$ if and only if the entropy of $p^*$ is bounded by

$$\mathbb{E}_{X \sim q}[\mathcal{L}(p^* \parallel \hat{p}; X)] - \varepsilon \leq H_{p^*}(Y \mid X) \leq \mathbb{E}_{X \sim q}[\mathcal{L}(p^* \parallel \hat{p}; X)]. \tag{8}$$

Therefore, if the model has low KL divergence, then entropy calibration can also be thought of as requiring that the model's entropy is close to the entropy of $p^*$ (Braverman et al., 2020).

(b) Due to the possibility of error accumulation during autoregressive generation, a model with low KL is not necessarily entropy calibrated. In particular, even for a model with only $\varepsilon$ KL divergence per time step, Corollary 4.2 of Braverman et al. (2020) shows that the entropy at the $t$-th step of generation can deviate as much as $\varepsilon + \sqrt{\varepsilon t}$ from that of $p^*$, growing with $t$.

(c) Like in binary calibration, one can easily attain entropy calibration by predicting the uniform distribution for all inputs. Therefore, a calibration guarantee is only meaningful if it is accompanied by a guarantee that model quality is preserved.

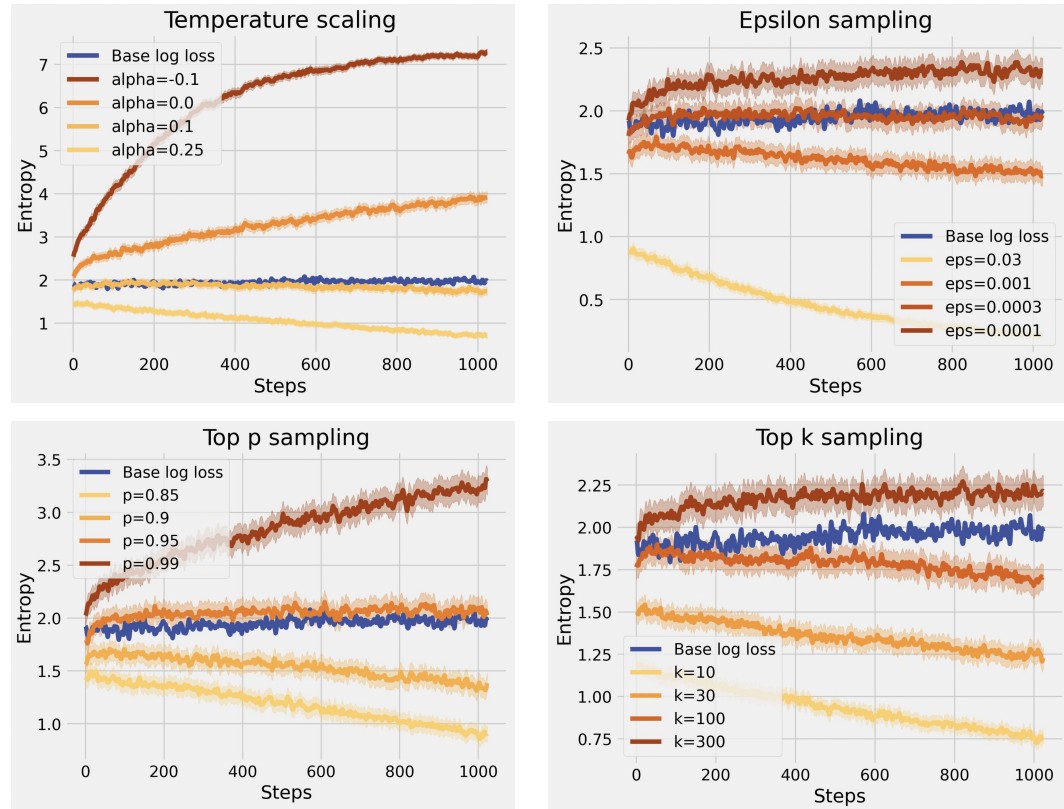

Figure 4: Generation entropy per time step of `TinyLlama_v1.1` applied to `wikitext-103` with various truncation techniques applied, compared to the unadjusted model's teacher-forced log loss (in blue). In each method (temperature scaling, epsilon sampling, top-p sampling, top-k sampling), the choice of truncation parameter shifts the entropy curve downward while also reducing the slope. The parameter choice that stabilizes the model is the one with slope close to zero.

## 4 MISCALIBRATION IN LANGUAGE MODELS

Empirically, entropy is a useful indicator of generation quality and diversity: entropy too high typically indicates that generations are too random and incoherent, while entropy too low indicates that generations have little variation. Therefore, models can be better understood by measuring their entropy calibration error, and sampling methods can be better understood in terms of how they affect miscalibration. With this insight, we find the following:

**Current language models are miscalibrated.** We first plot the entropy of a range of models, from `Llama-2-7B` to `Llama-2-70B` (Touvron et al., 2023), on the `writingprompts` dataset (Fan et al., 2018), where we give the models 128 tokens of context and ask it to generate 1024 additional tokens (Figure 1). We average over 1024 examples and use quantization to fit models in GPU memory (Dettmers et al., 2022); please see the appendix for other experimental details. For each model, we plot the entropy at each step of generation, and we compare these curves to the best model's log loss on actual human-written examples, which serves as an upper bound for the entropy of human text. In these plots, we observe the following:

(a) Base language models have entropy per step increasing over time, regardless of size: stronger models start with lower entropy but deviate upward at a similar rate as weaker models. Due to this deviation, generations become incoherent as more tokens are generated (see, e.g., Figure 2).

One explanation for this upward deviation is that because log loss severely penalizes putting zero probability on valid tokens, but only weakly penalizes putting non-zero probability on invalid tokens, language models are incentivized to put small amounts of probability on a large

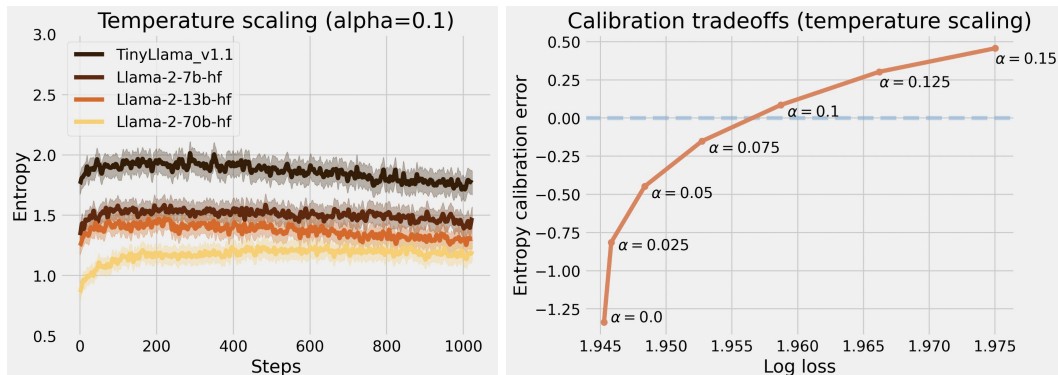

Figure 5: Left: the same temperature setting $\alpha = 0.1$, corresponding to temperature $0.909$, applied to all four base models on the `wikitext-103` dataset. Because models across different sizes are similarly miscalibrated, they are also best sampled at similar temperatures. Right: entropy calibration error plotted against log loss for various temperature settings, applied to `TinyLlama_v1.1` on the `wikitext-103` dataset. The unadjusted model attains the best log loss, and adjusting temperature improves calibration at the cost of increasing log loss.

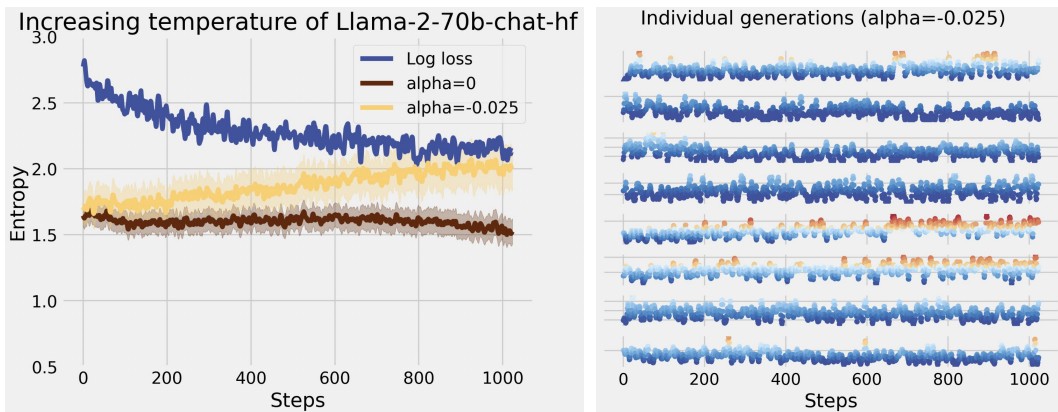

Figure 6: Left: Generation entropy per time step of `Llama-2-70b-chat-hf` applied to `writingprompts` with no temperature change ($\alpha = 0$) or a slight temperature increase ($\alpha = -0.025$, or temperature $1.026$), compared to the calibration target (in blue). Right: Entropy per time step for eight individual generations with temperature $1.026$ (blue: low entropy, red: high entropy). While we might hope to calibrate overconfident models by increasing the temperature, even a slight temperature increase causes entropy to become unstable, increasing over time on average. This increase is also not evenly distributed across generations: instead, individual generations become more volatile, with some generations remaining low entropy and others completely derailing.

number of both valid and invalid tokens (Hewitt et al., 2022). Also, models typically have high entropy on inputs containing invalid tokens. Then, the model's entropy will be higher for later generation steps, where it more likely that the prefix contains at least one invalid token.

(b) After instruction tuning, smaller models still have entropy too high, while larger models have entropy too low. This "overconfidence" of large instruction-tuned models is reflected in outputs lacking diversity and sometimes growing repetitive over time.

One explanation for this pattern is that instruction tuning encourages models to restrict to a subset of the language distribution, reducing entropy. Then, large models, which have larger capacity to overfit to the instruction tuning step, have lower entropy than smaller models.

If these trends continue, we expect that as model sizes grow, base models will continue to have entropy deviating upward, while instruction-tuned models will become more and more overconfident.

Diversity has been found to be especially important when solving difficult tasks that require picking from multiple generations (Li et al., 2022), generating synthetic data (Wang et al., 2023), or improving outputs by synthesizing multiple responses (Wang et al., 2024). Given that existing sampling methods are designed to decrease entropy rather than increase it, this situation suggests that we are in need of methods that calibrate overconfident models.

**Sampling parameters should be chosen to stabilize entropy.** In Figure 4, we plot the entropy per time step of `TinyLlama_v1.1` (Zhang et al., 2024) on `wikitext-103` (Merity et al., 2017) with various sampling techniques applied, including temperature scaling, epsilon sampling (Hewitt et al., 2022), top-p (nucleus) sampling (Holtzman et al., 2020), and top-k sampling (Fan et al., 2018). We find that for every method, adjusting the sampling parameter to make truncation more aggressive shifts the model's entropy downward and decreases the slope. If our goal is for entropy to be stable over time, we should then choose the parameter which adjusts the slope to be close to zero.

We then apply the most stable temperature setting for `TinyLlama_v1.1` ($\alpha = 0.1$, or temperature 0.909) to the larger Llama models (Figure 5). We find that because large and small models are similarly miscalibrated, the same temperature setting works well for all four models. The downside is that this stabilization comes at the cost of increased log loss due to reduced diversity, reproducing similar findings in past work (Hashimoto et al., 2019; Zhang et al., 2021; Pillutla et al., 2021).

For instruction-tuned models, on the other hand, which have too little entropy, one might be tempted to calibrate by increasing the temperature. While this approach can calibrate the model on average, it does so by causing some generations to derail upward while other generations remain low entropy (Figure 6). This degradation is not reflected in the log loss: log loss actually improves when increasing the temperature (from 2.29 to 2.28), due to the model originally having too little diversity. One approach in this setting might involve first increasing temperature to increase diversity, and then calibrating the entropy back down with a procedure that preserves diversity. Unfortunately, existing entropy reduction techniques do not preserve diversity.

## 5 FUTURE ENTROPY SCALING

### 5.1 THEORY

Because global adjustments are intractable and per-step adjustments increase log loss, a natural middle ground is an algorithm that makes per-step adjustments with some global information. This point of view motivates the *future-entropy-adjusted* model, which is given by

$$\hat{p}_\alpha^{\text{ent}}(y_t \mid x, y_{<t}) = \text{softmax}\{(1 + \alpha_t) \log \hat{p}(y_t \mid x, y_{<t}) - \alpha_t H_{\hat{p}_\alpha^{\text{ent}}}(\hat{Y}_{>t} \mid x, y_{<t}, y_t)\} \quad (9)$$

for calibration parameters $\alpha_1, ..., \alpha_T$, and where

$$H_{\hat{p}_\alpha^{\text{ent}}}(\hat{Y}_{>t} \mid x, y_{<t}, y_t) = \mathbb{E}_{\hat{Y}_{>t} \sim \hat{p}_\alpha^{\text{ent}}(\hat{Y}_{>t} \mid x, y_{<t}, y_t)}[-\log \hat{p}_\alpha^{\text{ent}}(\hat{Y}_{>t} \mid x, y_{<t}, y_t)] \quad (10)$$

denotes the total entropy of the entire continuation $\hat{Y}_{>t}$ if candidate token $y_t$ were to be chosen. Intuitively, a positive $\alpha$ corresponds to not only decreasing the temperature, but also penalizing tokens whose continuations have high entropy on average (and the reverse if $\alpha$ is negative). Our main result is that for this specific form of adjustment, for any initial model $\hat{p}$, one can simultaneously achieve calibration and improve log loss by choosing each $\alpha_t$ to minimize log loss:

$$\alpha_t^* = \underset{\alpha_t}{\text{argmin}} \, \mathbb{E}_{X \sim q}[\mathcal{L}_t(p^* \parallel \hat{p}_\alpha^{\text{ent}}; X)]. \quad (11)$$

Unfortunately, estimating the future entropy of $\hat{p}_\alpha$ is not tractable without further assumptions. One can estimate the entropy of $\hat{p}$ to $\varepsilon$ error by averaging over $O((T/\varepsilon^2) \log |V|)$ samples (Algorithm 2) because future entropy is bounded by $T \log |V|$, where $T$ is the length and $|V|$ is the vocab size. However, sampling exactly from $\hat{p}_\alpha^{\text{ent}}$ takes exponential time because evaluating $\hat{p}_\alpha^{\text{ent}}(\cdot \mid x, y_{<t})$ involves recursively evaluating $\hat{p}_\alpha^{\text{ent}}(\cdot \mid x, y_{<t}, y_t)$ for every candidate token $y_t \in V$. Therefore, we need to assume the existence of a surrogate model $\hat{p}'_\alpha$ whose future entropy approximates that of $\hat{p}_\alpha^{\text{ent}}$. With such a model, computing and sampling from $\hat{p}_\alpha^{\text{ent}}$ becomes tractable.

With this assumption, we prove the following result (please see the appendix for the full proof):

---

**Algorithm 1** Future entropy scaling

---

**Inputs:** model $\hat{p}$, max length $T$, future entropy estimator $\hat{H}(x, y_{<t}, y_t; \alpha_{>t})$, prompt distribution $q$, true conditional distribution $p^*$

1: Define

$$\hat{p}^{\text{ent}}(y_t \mid x, y_{<t}; \alpha_t, \alpha_{>t}) = \text{softmax}\{(1 + \alpha_t) \log \hat{p}(y_t \mid x, y_{<t}) - \alpha_t \hat{H}(x, y_{<t}, y_t; \alpha_{>t})\}.$$

2: Initialize $\alpha_1 = ... = \alpha_T = 0$.

3: For $t = T, ..., 1$:

4:      Choose $\alpha_t$ to minimize expected log loss at step $t$:

$$\alpha_t = \operatorname*{argmin}_{\alpha'_t} \mathbb{E}_{X \sim q}[\mathbb{E}_{Y \sim p^*(Y|X)}[-\log \hat{p}^{\text{ent}}(Y_t \mid X, Y_{<t}; \alpha'_t, \alpha_{>t})]].$$

5: Return $\alpha_1, ..., \alpha_T$.

---

**Algorithm 2** Future entropy estimation (sampling)

---

**Inputs:** surrogate model $\hat{p}'_\alpha$, max length $T$, prefix $z = (x, y_{<t}, y_t)$, number of samples $n$

1: Sample $n$ trajectories from the model applied to prefix $z$: $\left(\hat{Y}_{t+1}^{(i)}, ..., \hat{Y}_T^{(i)}\right)_{i=1}^n \overset{\text{i.i.d.}}{\sim} \hat{p}'_\alpha(\hat{Y}_{>t} \mid z)$.

2: Compute

$$\hat{H} = \frac{1}{n} \sum_{i=1}^n \sum_{s=t+1}^T -\log \hat{p}'_\alpha(\hat{Y}_s^{(i)} \mid z, \hat{Y}_{<s}^{(i)}).$$

3: Return $\hat{H}$.

---

**Theorem 5.1.** *Suppose that the future entropy estimator $\hat{H}$ satisfies $|\hat{H}(z; \alpha_{>t}) - H_{\hat{p}_\alpha^{\text{ent}}}(\hat{Y}_{>t} \mid z)| \leq \delta$ uniformly over prefixes $z$ and parameters $\alpha$. Then, the output of Algorithm 1 satisfies*

$$|EntCE(p^* \parallel \hat{p}_\alpha^{ent})| \leq T\delta,$$
$$\mathbb{E}_{X \sim q}[\mathcal{L}(p^* \parallel \hat{p}_\alpha^{ent}; X)] \leq \mathbb{E}_{X \sim q}[\mathcal{L}(p^* \parallel \hat{p}; X)].$$

*If each $\alpha_t$ is an $\varepsilon_t$-stationary point instead of an exact stationary point, then we instead have*

$$|EntCE(p^* \parallel \hat{p}_\alpha^{ent})| \leq T\delta + \sum_{t=1}^T (1 + \alpha_t)\varepsilon_t.$$

At a high level, the proof involves taking the gradient of the log loss with respect to each $\alpha_t$ and using the fact that it is small to show a certain calibration-like guarantee for each $t$. Combining these guarantees with induction then provides the full calibration guarantee.

### 5.2 EXPERIMENTS

While future entropy scaling provably preserves log loss, the most straightforward implementation involves averaging over multiple samples per candidate token, which is expensive (Algorithm 2). Nonetheless, we provide evidence that using future entropy is necessary empirically to avoid model degradation when calibrating, suggesting that efficient approximations of future entropy scaling can improve upon existing sampling techniques.

First, we plot the histogram of future entropy values for low probability tokens ($p < 0.0003$) and compare it to the histogram for high probability tokens ($p > 0.01$) (Figure 7). For 512 prefixes from `wikitext-103`, we estimate the 32-step future entropy (averaged over 32 trajectories) of the top 512 tokens of `TinyLlama_v1.1`. To interpret future entropy as an indicator for derailing, we define the baseline future entropy of a prefix as the average future entropy for high-probability tokens (which we assume are unlikely to derail the model). Then, for a given prefix, a token derails

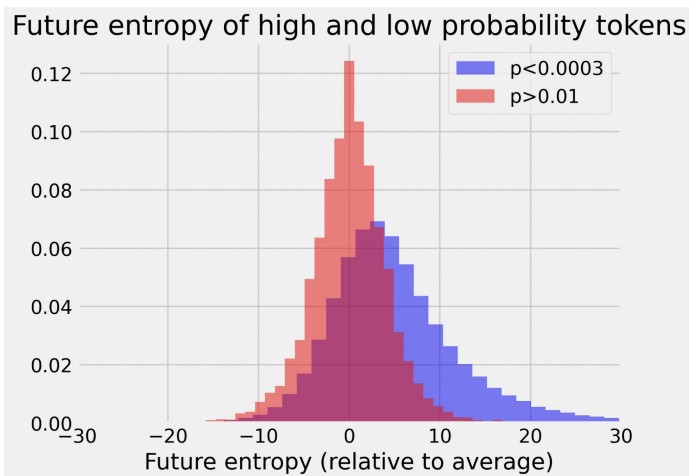

Figure 7: Histograms of the 32-step future entropy (relative to the average over high probability tokens for that prefix) for high probability tokens (in red) versus low probability tokens (in blue), for `TinyLlama_v1.1` applied to examples from `wikitext-103`. We find that there is substantial overlap between the two histograms, suggesting that there are many low-probability tokens that do not derail the generation, and some moderate-probability tokens that do derail the generation.

the model if it leads to a future entropy substantially larger than the baseline future entropy: models typically have high entropy when the input contains invalid tokens, leading to incoherent text.

In this plot, we find that there is substantial overlap between the two histograms: in other words, there are many low-probability tokens that do not derail the generation, and some tokens with moderate probability that do. Therefore, existing truncation algorithms, which only look at the token probabilities, cannot suppress tokens that cause derailing without also suppressing tokens that do not, leading to loss in diversity.

Next, to gain insight into why these histograms have so much overlap, we qualitatively examine `TinyLlama_v1.1` predictions on `wikitext-103`, and we find that future entropy is crucial for the following cases (see the appendix for examples):

(a) **Correcting model error:** The model sometimes assigns too much probability to incorrect continuations and too little probability to correct ones. In such cases, algorithms which only look at the next word probabilities, like temperature scaling, cannot suppress incorrect continuations without suppressing correct ones as well. Such examples suggest that future entropy lookahead is powerful enough to detect many model errors because errors often derail generation.

(b) **Avoiding tokens that increase generation difficulty:** In other cases, the model assigns moderate probability to a token that is valid but raises the difficulty of the subsequent generation. Figure 3 includes one such example where the model tasks its future self with generating a poem in Middle Scots; more examples are in the appendix. In these cases, future entropy serves the role of measuring prompt difficulty, helping the model avoid generating such prompts.

## 6 CONCLUSION

In this paper, we provided theory, algorithms, and analysis to better understand the entropy calibration of language models. Entropy miscalibration is a fundamental problem in autoregressive generation: theoretically, even very accurate models can have entropy deviating over time due to error accumulation, and empirically, large models are just as miscalibrated as smaller ones. Existing sampling methods, while beneficial, are myopic, hurt diversity, and are ill-suited for calibrating overconfident models. On the other hand, our analysis of future entropy scaling suggests calibration is possible without these tradeoffs. We hope that our work inspires new calibration techniques that improve the quality and diversity of language model generations.

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

# A  PROOFS

Recall: let $V$ denote the vocabulary, and let the prompt $X \in V^*$ and response $Y \in V^*$ be random variables taking values in $V^*$, the space of all strings over $V$. Also, let $X \sim q$ and $Y \sim p^*(Y \mid X)$ denote the ground truth prompt and response distributions, and let $\hat{p} : V^* \to \Delta^{|V|}$ be a language model mapping any string to a next token distribution over $V$. We will use $\hat{Y} \sim \hat{p}(\hat{Y} \mid X)$ to denote the response distribution induced by sampling autoregressively starting from the prompt $X$.

For a fixed prompt $X$, $\mathcal{L}(p^* \parallel \hat{p}; X)$ denotes the model's expected log loss on that prompt, and $H_{\hat{p}}(\hat{Y} \mid X)$ denotes the model's entropy on that prompt:

$$\mathcal{L}(p^* \parallel \hat{p}; X) = \mathbb{E}_{Y \sim p^*(Y|X)}[- \log \hat{p}(Y \mid X)]$$

$$= \mathbb{E}_{Y \sim p^*(Y|X)} \left[ \sum_{t=1}^{\text{len}(Y)} - \log \hat{p}(Y_t \mid X, Y_{<t}) \right]$$

$$H_{\hat{p}}(\hat{Y} \mid X) = \mathbb{E}_{\hat{Y} \sim \hat{p}(\hat{Y}|X)}[- \log \hat{p}(\hat{Y} \mid X)]$$

$$= \mathbb{E}_{\hat{Y} \sim \hat{p}(\hat{Y}|X)} \left[ \sum_{t=1}^{\text{len}(\hat{Y})} - \log \hat{p}(\hat{Y}_t \mid X, \hat{Y}_{<t}) \right].$$

Then, *entropy calibration error* is given by

$$\text{EntCE}(p^* \parallel \hat{p}) = \mathbb{E}_{X \sim q}[\mathcal{L}(p^* \parallel \hat{p}; X) - H_{\hat{p}}(\hat{Y} \mid X)].$$

Let the *future-entropy-adjusted* model be given by

$$\hat{p}_\alpha^{\text{ent}}(y_t \mid x, y_{<t}) = \text{softmax}\{(1 + \alpha_t) \log \hat{p}(y_t \mid x, y_{<t}) - \alpha_t H_{\hat{p}_\alpha^{\text{ent}}}(\hat{Y}_{>t} \mid x, y_{<t}, y_t)\}$$

for calibration parameters $\alpha_1, ..., \alpha_T$, and where

$$H_{\hat{p}_\alpha^{\text{ent}}}(\hat{Y}_{>t} \mid x, y_{<t}, y_t) = \mathbb{E}_{\hat{Y}_{>t} \sim \hat{p}_\alpha^{\text{ent}}(\hat{Y}_{>t}|x, y_{<t}, y_t)}[- \log \hat{p}_\alpha^{\text{ent}}(\hat{Y}_{>t} \mid x, y_{<t}, y_t)]$$

denotes the total entropy of the entire continuation $Y_{>t}$ if candidate token $y_t$ were to be chosen. Then, we have that

**Theorem A.1.** *Suppose that the future entropy estimator $\hat{H}$ satisfies $|\hat{H}(z; \alpha_{>t}) - H_{\hat{p}_\alpha^{\text{ent}}}(\hat{Y}_{>t} \mid z)| \leq \delta$ uniformly over prefixes $z$ and parameters $\alpha$. Then, the output of Algorithm 1 satisfies*

$$|EntCE(p^* \parallel \hat{p}_\alpha^{ent})| \leq 2T\delta,$$
$$\mathbb{E}_{X \sim q}[\mathcal{L}(p^* \parallel \hat{p}_\alpha^{ent}; X)] \leq \mathbb{E}_{X \sim q}[\mathcal{L}(p^* \parallel \hat{p}; X)].$$

*If each $\alpha_t$ is an $\varepsilon_t$-stationary point instead of an exact stationary point, then we instead have*

$$|EntCE(p^* \parallel \hat{p}_\alpha^{ent})| \leq 2T\delta + \sum_{t=1}^T (1 + \alpha_t)\varepsilon_t.$$

The proof proceeds as follows: first, we take the gradient of the log loss with respect to each $\alpha_t$ and use the fact that it is small to show a certain calibration-like guarantee for each $t$. We then combine these guarantees with induction to provide the full calibration guarantee.

**Lemma A.2.** *Under the setting of Theorem A.1, suppose that $\alpha_t$ is an $\varepsilon$-stationary point:*

$$\left| \frac{d}{d\alpha_t'} \mathbb{E}_{X \sim q}[\mathbb{E}_{Y \sim p^*(Y|X)}[- \log \hat{p}^{ent}(Y_t \mid X, Y_{<t}; \alpha_t', \alpha_{>t})]] \right| \leq \varepsilon.$$

*Then, we have the following bound:*

$$\left| \mathbb{E}_{X \sim q} \left[ \mathbb{E}_{\substack{Y_{\leq t} \sim p^*(Y_{\leq t}|X) \\ \hat{Y}_{>t} \sim \hat{p}_\alpha^{ent}(\hat{Y}_{>t}|X, Y_{\leq t})}} [- \log \hat{p}_\alpha^{ent}(Y_{\leq t}, \hat{Y}_{>t} \mid X)] \right. \right.$$

$$\left. \left. - \mathbb{E}_{\substack{Y_{<t} \sim p^*(Y_{<t}|X) \\ \hat{Y}_{\geq t} \sim \hat{p}_\alpha^{ent}(\hat{Y}_{\geq t}|X, Y_{<t})}} [- \log \hat{p}_\alpha^{ent}(Y_{<t}, \hat{Y}_{\geq t} \mid X)] \right] \right| \leq (1 + \alpha_t)\varepsilon + 2\delta.$$

This lemma provides us with a partial calibration guarantee in the sense that it lets us swap out $Y_t \sim p^*$ for $\hat{Y}_t \sim \hat{p}_\alpha^{\text{ent}}$ in the expectation. The next lemma is helpful in showing that the $t$-th iteration of Algorithm 1 preserves log loss:

**Lemma A.3.** *At the $t$-th iteration of Algorithm 1, let $\alpha_{t+1}, ..., \alpha_T$ be set arbitrarily, and let $\alpha_1, ..., \alpha_{t-1} = 0$. Then, we have*

$$\underset{\alpha_t'}{\arg\min} \, \mathbb{E}_{X \sim q}[\mathbb{E}_{Y \sim p^*(Y|X)}[-\log \hat{p}^{ent}(Y_t \mid X, Y_{<t}; \alpha_t', \alpha_{>t})]]$$

$$= \underset{\alpha_t'}{\arg\min} \, \mathbb{E}_{X \sim q}[\mathbb{E}_{Y \sim p^*(Y|X)}[-\log \hat{p}^{ent}(Y \mid X; \alpha_{<t}, \alpha_t', \alpha_{>t})]];$$

*in other words, optimizing $\alpha_t$ with respect to the log loss at time $t$ is equivalent to optimizing $\alpha_t$ with respect to the full log loss over all time steps.*

Combining these guarantees for $t = 1, ..., T$ then provides a full calibration guarantee:

*Proof of Theorem A.1.* We will prove the calibration bound by induction. Applying Lemma A.2 for $t = 1$, we have

$$\left| \mathbb{E}_{X \sim q} \left[ \mathbb{E}_{\substack{Y_1 \sim p^*(Y_1|X) \\ \hat{Y}_{2,...,T} \sim \hat{p}_\alpha^{\text{ent}}(\hat{Y}_{2,...,T}|X,Y_1)}} [-\log \hat{p}_\alpha^{\text{ent}}(Y_1, \hat{Y}_{2,...,T} \mid X)] \right. \right.$$

$$\left. \left. - \mathbb{E}_{\hat{Y}_{1,...,T} \sim \hat{p}_\alpha^{\text{ent}}(\hat{Y}_{1,...,T}|X)}[-\log \hat{p}_\alpha^{\text{ent}}(\hat{Y}_{1,...,T} \mid X)] \right] \right| \le (1 + \alpha_1)\varepsilon_1 + 2\delta.$$

For ease of notation, we will write this guarantee as

$$|\tilde{H}(\{1\}, \{2, ..., T\}) - \tilde{H}(\{\}, \{1, ..., T\})| \le (1 + \alpha_1)\varepsilon_1 + 2\delta$$

for $\tilde{H}(I, J)$ given by

$$\tilde{H}(I, J) = \mathbb{E}_{X \sim q} \left[ \mathbb{E}_{\substack{Y_I \sim p^*(Y_I|X) \\ \hat{Y}_J \sim \hat{p}_\alpha^{\text{ent}}(\hat{Y}_J|X,Y_I)}} [-\log \hat{p}_\alpha^{\text{ent}}(Y_I, \hat{Y}_J \mid X)] \right].$$

As our inductive hypothesis, suppose that for time $t$, we have that

$$|\tilde{H}(\{1, ..., t\}, \{t+1, ..., T\}) - \tilde{H}(\{\}, \{1, ..., T\})| \le 2t\delta + \sum_{s=1}^{t} (1 + \alpha_s)\varepsilon_s.$$

By Lemma A.2 for $t + 1$, we have

$$|\tilde{H}(\{1, ..., t\}, \{t+1, ..., T\}) - \tilde{H}(\{1, ..., t+1\}, \{t+2, ..., T\})| \le (1 + \alpha_{t+1})\varepsilon_{t+1} + 2\delta.$$

Then, applying the triangle inequality, we have

$$|\tilde{H}(\{1, ..., t+1\}, \{t+2, ..., T\}) - \tilde{H}(\{\}, \{1, ..., T\})| \le 2(t+1)\delta + \sum_{s=1}^{t+1} (1 + \alpha_s)\varepsilon_s,$$

completing the inductive step.

To show that log loss is preserved, let $\alpha = (\alpha_1, ..., \alpha_T)$ be output of the algorithm, and let $\alpha^t = (0, ..., 0, \alpha_t, ..., \alpha_T)$ be the setting of $\alpha$ after the $t$-th iteration for $t = T, ..., 1$. By Lemma A.3 applied to iteration $t$, we have that

$$\mathbb{E}_{X \sim q}[\mathbb{E}_{Y \sim p^*(Y|X)}[-\log \hat{p}_{\alpha^t}^{\text{ent}}(Y \mid X)]] \le \mathbb{E}_{X \sim q}[\mathbb{E}_{Y \sim p^*(Y|X)}[-\log \hat{p}_{\alpha^{t+1}}^{\text{ent}}(Y \mid X)]],$$

where we define $\alpha^{T+1} = (0, ..., 0)$ (so $\hat{p}_{\alpha^{T+1}}^{\text{ent}} = \hat{p}$), because each $\alpha_t$ is chosen to minimize log loss. Because log loss improves at every step, we then have that

$$\mathbb{E}_{X \sim q}[\mathbb{E}_{Y \sim p^*(Y|X)}[-\log \hat{p}_{\alpha^1}^{\text{ent}}(Y \mid X)]] \le \mathbb{E}_{X \sim q}[\mathbb{E}_{Y \sim p^*(Y|X)}[-\log \hat{p}_{\alpha^{T+1}}^{\text{ent}}(Y \mid X)]]$$

as desired. $\qquad\square$

It remains to prove the two lemmas, which we do below:

*Proof of Lemma A.2.* Taking the derivative of log loss with respect to $\alpha_t$, we have

$$\varepsilon \geq \frac{d}{d\alpha_t}\mathbb{E}_{X\sim q}[\mathbb{E}_{Y\sim p^*(Y|X)}[-\log\hat{p}^{\text{ent}}(Y_t \mid X, Y_{<t}; \alpha_t, \alpha_{>t})]]$$

$$= \frac{d}{d\alpha_t}\mathbb{E}_{X\sim q}[\mathbb{E}_{Y\sim p^*(Y|X)}[-\log\text{softmax}((1+\alpha_t)\log\hat{p}(Y_t \mid X, Y_{<t}) - \alpha_t\hat{H}(X, Y_{<t}, Y_t; \alpha_{>t}))]]$$

$$= \mathbb{E}_{X\sim q}[\mathbb{E}_{Y\sim p^*(Y|X)}[-(\mathbb{1}_{Y_t} - \hat{p}^{\text{ent}}_\alpha(\cdot \mid X, Y_{<t}))^T(\log\hat{p}(\cdot \mid X, Y_{<t}) - \hat{H}(X, Y_{<t}, \cdot; \alpha_{>t}))]]$$

$$= \mathbb{E}_{X\sim q}[\mathbb{E}_{Y_{\leq t}\sim p^*(Y_{\leq t}|X)}[-(\log\hat{p}(Y_t \mid X, Y_{<t}) - \hat{H}(X, Y_{<t}, Y_t; \alpha_{>t}))]]$$

$$- \mathbb{E}_{X\sim q}\left[\mathbb{E}_{\substack{Y_{<t}\sim p^*(Y_{<t}|X)\\ \hat{Y}_t\sim p^{\text{ent}}_\alpha(\hat{Y}_t|X,Y_{<t})}}\left[-(\log\hat{p}(\hat{Y}_t \mid X, Y_{<t}) - \hat{H}(X, Y_{<t}, \hat{Y}_t; \alpha_{>t}))\right]\right],$$

where the two terms only differ in whether $Y_t \sim p^*$ or $\hat{Y}_t \sim \hat{p}^{\text{ent}}_\alpha$. Next, we can multiply both sides by $(1+\alpha_t)$ to get

$$(1+\alpha_t)\varepsilon$$

$$\geq \mathbb{E}_{X\sim q}[\mathbb{E}_{Y_{\leq t}\sim p^*(Y_{\leq t}|X)}[-((1+\alpha_t)\log\hat{p}(Y_t \mid X, Y_{<t}) - (1+\alpha_t)\hat{H}(X, Y_{<t}, Y_t; \alpha_{>t}))]]$$

$$- \mathbb{E}_{X\sim q}\left[\mathbb{E}_{\substack{Y_{<t}\sim p^*(Y_{<t}|X)\\ \hat{Y}_t\sim p^{\text{ent}}_\alpha(\hat{Y}_t|X,Y_{<t})}}\left[-((1+\alpha_t)\log\hat{p}(\hat{Y}_t \mid X, Y_{<t}) - (1+\alpha_t)\hat{H}(X, Y_{<t}, \hat{Y}_t; \alpha_{>t}))\right]\right].$$

Note that these expressions look similar to the argument of the softmax in the definition of $\hat{p}^{\text{ent}}_\alpha$, with only $Y_t$ differing from $\hat{Y}_t$. Both expressions are only missing the same normalizing constant, so we can add and subtract this normalizing constant to get

$$= \mathbb{E}_{X\sim q}[\mathbb{E}_{Y_{\leq t}\sim p^*(Y_{\leq t}|X)}[-(\log\hat{p}^{\text{ent}}_\alpha(Y_t \mid X, Y_{<t}) - \hat{H}(X, Y_{<t}, Y_t; \alpha_{>t}))]]$$

$$- \mathbb{E}_{X\sim q}\left[\mathbb{E}_{\substack{Y_{<t}\sim p^*(Y_{<t}|X)\\ \hat{Y}_t\sim p^{\text{ent}}_\alpha(\hat{Y}_t|X,Y_{<t})}}\left[-(\log\hat{p}^{\text{ent}}_\alpha(\hat{Y}_t \mid X, Y_{<t}) - \hat{H}(X, Y_{<t}, \hat{Y}_t; \alpha_{>t}))\right]\right].$$

Next, we can add and subtract $\mathbb{E}_{X\sim q}\mathbb{E}_{Y_{<t}\sim p^*(Y_{<t}|X)}[-\log\hat{p}^{\text{ent}}_\alpha(Y_{<t} \mid X)]$ from the right hand side to get

$$= \mathbb{E}_{X\sim q}[\mathbb{E}_{Y_{\leq t}\sim p^*(Y_{\leq t}|X)}[-(\log\hat{p}^{\text{ent}}_\alpha(Y_{<t}, Y_t \mid X) - \hat{H}(X, Y_{<t}, Y_t; \alpha_{>t}))]]$$

$$- \mathbb{E}_{X\sim q}\left[\mathbb{E}_{\substack{Y_{<t}\sim p^*(Y_{<t}|X)\\ \hat{Y}_t\sim p^{\text{ent}}_\alpha(\hat{Y}_t|X,Y_{<t})}}\left[-(\log\hat{p}^{\text{ent}}_\alpha(Y_{<t}, \hat{Y}_t \mid X) - \hat{H}(X, Y_{<t}, \hat{Y}_t; \alpha_{>t}))\right]\right].$$

At this point, we can use the fact that $\hat{H}(X, Y_{<t}, \hat{Y}_t; \alpha_{>t})$ is within $\delta$ of the actual future entropy to get

$$(1+\alpha_t)\varepsilon + 2\delta \geq \mathbb{E}_{X\sim q}[\mathbb{E}_{Y_{\leq t}\sim p^*(Y_{\leq t}|X)}[-(\log\hat{p}^{\text{ent}}_\alpha(Y_{<t}, Y_t \mid X) - H_{p^{\text{ent}}_\alpha}(\hat{Y}_{>t} \mid X, Y_{<t}, Y_t))]]$$

$$- \mathbb{E}_{X\sim q}\left[\mathbb{E}_{\substack{Y_{<t}\sim p^*(Y_{<t}|X)\\ \hat{Y}_t\sim p^{\text{ent}}_\alpha(\hat{Y}_t|X,Y_{<t})}}\left[-(\log\hat{p}^{\text{ent}}_\alpha(Y_{<t}, \hat{Y}_t \mid X) - H_{p^{\text{ent}}_\alpha}(\hat{Y}_{>t} \mid X, Y_{<t}, Y_t))\right]\right].$$

Finally, note that by definition, we have

$$H_{p^{\text{ent}}_\alpha}(\hat{Y}_{>t} \mid X, Y_{<t}, Y_t)) = \mathbb{E}_{\hat{Y}_{>t}\sim\hat{p}^{\text{ent}}_\alpha(\hat{Y}_{>t}|X,Y_{<t},Y_t)}[-\log\hat{p}^{\text{ent}}_\alpha(\hat{Y}_{>t} \mid X, Y_{<t}, Y_t)],$$

which we can substitute into the previous equation to get

$$(1+\alpha_t)\varepsilon + 2\delta \geq \mathbb{E}_{X\sim q}\left[\mathbb{E}_{\substack{Y_{\leq t}\sim p^*(Y_{\leq t}|X)\\ \hat{Y}_{>t}\sim\hat{p}^{\text{ent}}_\alpha(\hat{Y}_{>t}|X,Y_{<t},Y_t)}}[-\log\hat{p}^{\text{ent}}_\alpha(Y_{<t}, Y_t, \hat{Y}_{>t} \mid X)]\right]$$

$$- \mathbb{E}_{X\sim q}\left[\mathbb{E}_{\substack{Y_{<t}\sim p^*(Y_{<t}|X)\\ \hat{Y}_{\geq t}\sim\hat{p}^{\text{ent}}_\alpha(\hat{Y}_{\geq t}|X,Y_{<t})}}[-\log\hat{p}^{\text{ent}}_\alpha(Y_{<t}, \hat{Y}_t, \hat{Y}_{>t} \mid X)]\right],$$

which proves the desired result.

$\square$

*Proof of Lemma A.3.* Let $t_0$ denote the time step of interest. We can first write the full log loss as a sum over $t$:

$$\mathbb{E}_{X \sim q}[\mathbb{E}_{Y \sim p^*(Y|X)}[-\log \hat{p}^{\text{ent}}(Y \mid X; \alpha)]]$$

$$= \sum_{t=1}^{T} \mathbb{E}_{X \sim q}[\mathbb{E}_{Y \sim p^*(Y|X)}[-\log \hat{p}^{\text{ent}}(Y_t \mid X, Y_{<t}; \alpha_t, \alpha_{>t})]].$$

Because $\alpha_{<t}$ has no involvement in the $t$-th prediction by the definition of future entropy scaling, we can remove the summands $t_0 + 1, ..., T$, which are constant with respect to $\alpha_{t_0}$. Next, note that $\alpha_1 = ... = \alpha_{t_0-1} = 0$, so the predictions for these time steps are not adjusted:

$$\hat{p}^{\text{ent}}(Y_t \mid X, Y_{<t}; 0, \alpha_{>t}) = \hat{p}(Y_t \mid X, Y_{<t}) \text{ for } t < t_0.$$

Therefore, all terms in the sum except the $t_0$th one are constant with respect to $\alpha_{t_0}$, proving the desired result. $\square$

## B  EXPERIMENTAL DETAILS

We use the TinyLlama (Zhang et al., 2024) and Llama 2 (Touvron et al., 2023) models (7b, 13b, 70b, 7b-chat, 13b-chat, 70b-chat) on the wikitext-103 (Merity et al., 2017) and writingprompts (Fan et al., 2018) datasets, in pytorch (Paszke et al., 2019) and Hugging Face transformers (Wolf et al., 2020). We use the xformers attention kernel (Lefaudeux et al., 2022), and models are quantized to 4 bits with bitsandbytes (Dettmers et al., 2022). Plots are generated in matplotlib (Hunter, 2007). To generate multiple continuations for a prefix to estimate future entropy, we use the attention masking trick described in Section 4.2 of Zelikman et al. (2024) to generate in parallel. All experiments are run on a NVIDIA RTX 6000 Ada Generation 49.1GB GPU.

## C  FUTURE ENTROPY EXAMPLES

Below, we provide examples from `TinyLlama_v1.1` applied to `wikitext-103`. Specifically, we compute the 64- or 128-step future entropy for the top 32 next tokens for each prefix, by averaging over 32 trajectories sampled with temperature 0.909. We then identify examples where the $\alpha = 0.1$ temperature adjustment differs substantially from the $\alpha = 0.1$ future entropy adjustment. We identify the following categories:

(a) **Model errors**: the model often assigns moderate probability to incorrect continuations. Many of these errors are due to choosing an alternate tokenization, inducing sudden topic shifts, choosing tokens that only work in other contexts, or assigning too much or too little probability to ellipses or newline characters. Some prefixes are also more difficult than others. As a result of model errors, temperature scaling must truncate valid tokens with low probability if it also wants to truncate invalid ones with moderate probability.

(b) **Increasing generation difficulty**: in other cases, the model assigns high probability to continuations that are valid but make derailing more likely in the future. Some cases include tokens that induce creative writing, or tokens that threaten a sudden topic change if not handled correctly. Lookahead is necessary to detect these cases and avoid generating them.

Examples are provided below ($\hat{p}$: original probability, $\hat{H}$: estimate of future entropy, $\hat{H}_{\text{avg}}$: average future entropy for the top 32 tokens, $\hat{p}_\alpha^{\text{temp}}$: probability after temperature scaling, $\hat{p}_\alpha^{\text{ent}}$: probability after future entropy scaling, $H$: entropy of the given continuation):

| Prompt | Continuations | Explanation |
|---|---|---|
| = Hello Good Morning = "Hello Good Morning" is a song by American rapper and producer Diddy and his band Dirty Money, from their debut album, Last Train to Paris. It was released from March 30, 2010 as the album's third single. The electronic dance song incorporates an acid squelch section in the middle 8, ad was written by Marcella Araica, Richard Butler, Clifford "T.I." Harris and Nathaniel "Danja" Hills who also produced the song. T.I. has a featured rap on the song. The song's | **Token:** main ($\hat{p} = 0.013$) 
 $\hat{H} = 141, \hat{H}_{\text{avg}} = 126$ 
 $\hat{p}_\alpha^{\text{temp}} = 0.010, \hat{p}_\alpha^{\text{ent}} = 0.0006$ 

 **Continuation** ($H = 193$): subject is "the past of and/or coming from a relationship and/or personal experience" focused around "the older sibling who has been there but ain't around anymore" on which Diddy sings, "The bruised, dirty, busted, broken / The come up after the coke | The token "main" causes the model to start writing about the subject and lyrics of the song, and the model is not strong enough to do so coherently. |
| = Clavaria zollingeri = Clavaria zollingeri, commonly known as the violet coral or the magenta coral, is a widely distributed species of fungus. It produces striking tubular, purple to pinkish-violet fruit bodies that grow up to 10 cm (3.9 in) tall and 7 cm (2.8 in) wide. The extreme tips of the fragile, slender branches are usually rounded and brownish. A typical member of the clavarioid or club fungi, Clavaria zollingeri is | **Token:** character ($\hat{p} = 0.016$) 
 $\hat{H} = 297, \hat{H}_{\text{avg}} = 274$ 
 $\hat{p}_\alpha^{\text{temp}} = 0.014, \hat{p}_\alpha^{\text{ent}} = 0.0005$ 

 **Continuation** ($H = 340$): ized by a fruticose coralstratified habit, alternating scales with at first green, but later yellow and tan, usually insuffers a perineal fungation on its inedible fleshy frond-like rhizoid. A Menzies suggested an origin of its species name, from the Latin name of the plant, tardifera, which means "slow-growing"... | In this example, choosing the token "character" forces the model to characterize a type of coral that it is not knowledgeable about, causing it to derail. |

| Prompt | Continuations | Explanation |
|---|---|---|
| = Directed acyclic graph =
In mathematics and computer science, a directed acyclic graph (DAG / ˈdæg /), is a finite directed graph with no directed cycles. That is, it consists of finitely many vertices and edges, with each edge directed from one vertex to another, such that there is no way to start at any vertex v and follow a consistently-directed sequence of edges that eventually loops back to v again. Equivalently, a DAG is a directed graph that has a topological ordering, a sequence of the vertices such that every edge is directed from earlier to later in the sequence. | **Token:** D ($\hat{p} = 0.04$)
$\hat{H} = 151, \hat{H}_{\text{avg}} = 124$
$\hat{p}_\alpha^{\text{temp}} = 0.038, \hat{p}_\alpha^{\text{ent}} = 0.0018$

**Continuation** ($H = 170$):
ifferentiation from acyclicity
Different concepts imply same knowledge.
Both are correct. But in the acyclic vs path acyclic paper, the pole types are used.
Acyclic refers to direction, not path way, given no self loop.
In path acyclic graph | In this example, the model assigns moderate probability to both "DA" ($\hat{p} = 0.06, \hat{H} = 127$) and "D" ($\hat{p} = 0.04, \hat{H} = 151$), but it has only seen "DAG" tokenized as "DA-G." Therefore, when it chooses the alternate tokenization "D," it is unable to generate "DAG" and derails as a result. |
| = U.S. Route 50 in Utah =
U.S. Route 50 (US-50) in Utah crosses the center of the state. The highway serves no major population centers in Utah, with the largest city along its path being Delta. Most of the route passes through desolate, remote areas. Through the eastern half of the state the route is concurrent with Interstate 70 (I-70). US-50 both enters and exits Utah concurrent with US-6, however the two routes are separate through the center of the state. | **Token:** U ($\hat{p} = 0.22$)
$\hat{H} = 132, \hat{H}_{\text{avg}} = 115$
$\hat{p}_\alpha^{\text{temp}} = 0.24, \hat{p}_\alpha^{\text{ent}} = 0.036$

**Continuation** ($H = 205$):
tub
Cleared land
From the Warburton Mine at Us-190, 44 hours of drive time (US 50), including a gap in the middle.
Allegheny Mountains;
Westwind,
Olivinus,
D: 03: | Like the example above, the model assigns high probability to "US" ($\hat{p} = 0.32, \hat{H} = 111$), the correct tokenization, and "U" ($\hat{p} = 0.24, \hat{H} = 132$), the incorrect tokenization. Because it has only seen "US" tokenized as one unit, it does not generate "S" after "U" and derails as a result. |
| = Jim and Mary McCartney =
James "Jim" McCartney (7 July 1902 – 18 March 1976) and Mary Patricia McCartney (née Mohan) (29 September 1909 – 31 October 1956) were the parents of musician, author and artist Paul McCartney of the Beatles and Wings, and younger brother photographer and musician Mike McCartney (better known professionally as Mike McGear), who worked with the comedy rock trio the Scaffold. | **Token:** I ($\hat{p} = 0.014$)
$\hat{H} = 153, \hat{H}_{\text{avg}} = 130$
$\hat{p}_\alpha^{\text{temp}} = 0.012, \hat{p}_\alpha^{\text{ent}} = 0.0004$

**Continuation** ($H = 186$):
was looking for an article entitiled "Sheba McCarthy makes her family proud " on Allen Maddox's site.
Found it and am still wondering what is he afraid of being truthful
Want to know what is he afraid of being truthful?
He can't read or write | In this example, the token "I" derails the generation by suddenly changing the tone from a third person article to first person dialogue. Nonetheless, the model still puts moderate probability on this token. |

| Prompt | Continuations | Explanation |
|---|---|---|
| = Black-tailed jackrabbit = The black-tailed jackrabbit (Lepus californicus), also known as the American desert hare, is a common hare of the western United States and Mexico, where it is found at elevations from sea level up to 10,000 ft (3,000 m). Reaching a length around 2 ft (61 cm), and a weight from 3 to 6 lb (1.4 to 2.7 kg), the black-tailed jackrabbit is the third-largest | **Token:** \n ($\hat{p} = 0.015$) $\hat{H} = 140, \hat{H}_{\text{avg}} = 119$ $\hat{p}_\alpha^{\text{temp}} = 0.013, \hat{p}_\alpha^{\text{ent}} = 0.0014$ 

 **Continuation** ($H = 188$): Raw Dog Food Used In The Jungle I have researched many things about this skin condition and have found or have been told, many things that are not correct. The skin condition I am afraid of is Eczema. It is a named dermatitic condition and can start very young and never end | The model often assigns moderate probability to the newline token despite being in the middle of a sentence. When the newline token is chosen in this way, the generation derails. |
| = Harajuku Lovers Tour = The Harajuku Lovers Tour was the first solo concert tour of American recording artist Gwen Stefani. The tour began through October to November 2005, to support of her debut studio album Love. Angel. Music. Baby. (2004). Although Stefani embarked on multiple tours with her band No Doubt, she initially opted not to participate in a tour to promote her album, an attitude that the singer eventually abandoned due to the commercial success of Love. Angel. Music. Baby. The Harajuku Lovers Tour | **Token:** \n ($\hat{p} = 0.012$) $\hat{H} = 75, \hat{H}_{\text{avg}} = 116$ $\hat{p}_\alpha^{\text{temp}} = 0.0098, \hat{p}_\alpha^{\text{ent}} = 0.12$ 

 **Continuation** ($H = 72$): The Harajuku Lovers Tour was the second solo concert tour of American recording artist Gwen Stefani. The tour kicked off in San Francisco, California, and ended in Los Angeles, California, continuing through the south of the United States from mid-April to mid-May. On March 1 | In contrast with the previous example, the newline token does not always derail the generation. Using lookahead enables the model to detect when the character should be truncated and when it should not. |
| = Stanley Matthews = Sir Stanley Matthews, CBE (1 February 1915 – 23 February 2000) was an English footballer. Often regarded as one of the greatest players of the English game, he is the only player to have been knighted while still playing, as well as being the first winner of both the European Footballer of the Year and the Football Writers' Association Footballer of the Year awards. Matthews' nicknames included "The Wizard of the Dribble" and "The Magician". Matthews kept | **Token:** pace ($\hat{p} = 0.008$) $\hat{H} = 148, \hat{H}_{\text{avg}} = 121$ $\hat{p}_\alpha^{\text{temp}} = 0.006, \hat{p}_\alpha^{\text{ent}} = 0.0002$ 

 **Continuation** ($H = 174$): with real-life speedsters like Billy Welsh, as well as other invented speed figures (including Panama Lincoln, Shaqiri), and had over 150 shots in a game against Huddersfield. A promising young player, he was accused by the press and his own club of | In this example, the model assigns moderate probability to "pace," which is a reasonable continuation to "kept" in other contexts but not in this one. Lookahead allows us to detect that this continuation is invalid and leads to derailing. |

| Prompt | Continuations | Explanation |
|---|---|---|
| = Allah =
Allah ( ... ) is the Arabic word referring to God in Abrahamic religions. The word is thought to be derived by contraction from al ilāh, which means "the God", and has cognates in other Semitic languages, including Elah in Aramaic, 'Ēl in Canaanite and Elohim in Hebrew. | **Token:** S ($\hat{p} = 0.014$)
$\hat{H} = 162, \hat{H}_{\text{avg}} = 141$
$\hat{p}_\alpha^{\text{temp}} = 0.012, \hat{p}_\alpha^{\text{ent}} = 0.0007$

**Continuation 1** ($H = 208$):
acred Secret Of Other Galaxies Unique Mechanisms Of Evolution Carbon Recycling In The Ocean Worksheets Science And Faith Coloreado Overswendner Home Bible Verses Books Of The Bible What Is The Owner Of The Seventh Seal Still On Earth Ancient Bee

**Continuation 2** ($H = 154$):
uch universal terms (in Arabic, either of two Arabic words [...], or of two Canaanite words [...]) are frequently used | In this example, the model generates the token "S," which makes derailing more likely in the future because only a few tokens (like "Such") stay on topic, while others lead to derailing. |
| = Orval Grove =
Orval Leroy Grove (August 29, 1919 – April 20, 1992) was an American pitcher in Major League Baseball who played for ten seasons in the American League with the Chicago White Sox. In 207 career games, Grove pitched 1,176 innings and posted a win – loss record of 63 – 73, with 66 complete games, 11 shutouts, and a 3.78 earned run average (ERA). The | **Token:** best ($\hat{p} = 0.018$)
$\hat{H} = 140, \hat{H}_{\text{avg}} = 108$
$\hat{p}_\alpha^{\text{temp}} = 0.016, \hat{p}_\alpha^{\text{ent}} = 0.0002$

**Continuation 1** ($H = 183$):
ones are those you devour whole and savor. - John Prine Whoever had to write a death poem or life quote always had to be a bit insecure. - Harold Pinter You've got to be outrageous in order to be true. - Michael Krasny Gover

**Continuation 2** ($H = 70$):
seasons in Grove's career came in 1947, when he won 19 games, was fourth in the league with a 2.73 ERA and had a career-best 184 strikeouts in 197 ⁄3 innings pitched over | Like the example above, choosing the token "best" makes derailing more likely because the subsequent token has the possibility of causing a sudden change in topic. |

