# OpenReview forum: "On the Entropy Calibration of Language Models"
_ICLR.cc/2025/Conference — ICLR 2025 Conference Withdrawn Submission_

### Official Review · Reviewer_fgRt · 2024-10-30

**Soundness:** 2
**Presentation:** 3
**Contribution:** 2
**Rating:** 5
**Confidence:** 3

**Summary:**

This paper presents an analysis of the entropy of the text generated by pre-trained language models, observing that (1) for base models, entropy rate increases as more tokens are generated (a result that echoes Braverman et al., 2020), and (2) after instruction tuning, the entropy of the generated text tends to decrease as model size increases, with a LLaMA 2 70B displaying very little entropy, compared to a baseline representing the entropy upper bound of human text.
To address this miscalibration, the authors propose future entropy scaling, a decoding approach that down-weights the probability assigned to tokens whose continuations have high entropy on average. This procedure relies on the assumption that the future entropy of the output text decoded using future entropy scaling can be approximated by the entropy of the generated text sampling from the unadjusted model output distribution.
The authors theoretically prove that future entropy scaling guarantees a bound on the entropy calibration error without increasing the model’s log loss.

**Strengths:**

- The paper presents an insightful analysis of entropy in text generated by language models across various model sizes and decoding strategies. The trend observed in instruction-tuned models is interesting.
- The proposed method effectively bounds the entropy of generated text while preserving log loss.
- The paper is well written and the contribution is clearly presented.

**Weaknesses:**

- Experiments are carried out on a single family of models. While the results might still be insightful, there are still some doubts about how generalizable the observations are. For instance, the statement “larger models now have too little entropy” (lines 89-90) is actually based on a result obtained with a single model (LLaMA 2 70B).
- In Section 4, the authors show that using temperature scaling to calibrate base models can effectively stabilize the entropy of the output, but comes at the cost of increased log loss. However, it’s unclear how significant the increase in log loss shown in Figure 6 is. Can the authors elaborate on the log loss/calibration trade-off and contextualize the numbers displayed in Figure 6?
- An underlying assumption behind the proposed method seems to be that tokens that lead to an increase in the entropy of the output are *always* undesired. Can there be cases in which higher entropy in a portion of the output might not necessarily be something negative? Related to this point, in Section 5.2, the authors mention that “future entropy is crucial for… Avoiding tokens that increase generation difficulty.” What does it mean for a generation to be “difficult”? This should be further discussed. For instance, is a sentence that contains a richer and more diverse vocabulary “more difficult” than one containing common and repeated words? Then a "more difficult" generation might not necessarily be undesirable.
- The proposed future entropy scaling implementation is computationally demanding, requiring multiple samples per candidate token. While the authors acknowledge this (line 477), further details on the computational cost and practical implementation challenges would strengthen the paper.

**Questions:**

I included my question in the "Weaknesses" section.

---

### Official Review · Reviewer_auj9 · 2024-11-03

**Soundness:** 3
**Presentation:** 4
**Contribution:** 2
**Rating:** 3
**Confidence:** 3

**Summary:**

This paper studies entropy calibration of pretrained and preference-optimized LLM. It reports that pretrained and smallish instruction-tuned LLMs have too high entropy, while a large instruction-tuned LLM has too low entropy. The paper analyzes some existing ways to tune the calibration, and finds them lacking. Finally, it proposes a novel way -- future entropy scaling -- to do calibration-tuning, and includes some preliminary results that show its promise.

**Strengths:**

This is an interesting and well-written paper on an important topic. It explains well the concept of entropy calibration, reports empirical results with 6 LLMs to illustrate the lack of calibration, explains well the shortcomings of existing calibration tuning techniques, and makes an interesting proposal on a new tuning technique.

**Weaknesses:**

The main weakness of the paper is that the results are all still quite preliminary, and need to be expanded on for this paper to be at the level that I think is required for an ICLR publication.

Figure 1b shows 1 instruction-tuned LLM that shows a different pattern based on a single prompt. This is intriguing, but the finding needs to be substantiated with more large instruction-tuned LLMs and more prompts.

Figure 2 shows a entropy profiles for 4 generations, and gives the generated text to illustrate that the high-entropy generation involves incoherent text ("derails"). Also in figure 6, and throughout the paper, loose claims are made that high entropy sections are incoherent. This is interesting, but I would really need to see much more solid evidence: this finding should be corroborated with many more generations and independent annotator results to judge the incoherence of the text.

Similarly, throughout the paper, including near figure 6 and at line 398, the claim is made that high entropy tokens early on *cause* incoherent generations later on. Interesting, but such a claim needs to be supported with statistical evidence.

Finally, the proposal of 'future entropy scaling' is interesting, but the paper really needs some empirical results showing that this, indeed, leads to better calibrated (and better perfoming?) LLMs.

In short, the paper's claims are interesting, but need to be much better supported with evidence. There now is quite a bit of redundancy in the text: the introduction summarizes all the findings. If that is shortened, there is also plentry of space left to discuss the additional results. I'd recommend doing the extra work, and submitting a much more convincing paper to an ML or NLP conference a bit later in 2025.

**Questions:**

No other questions or comments.

---

### Official Review · Reviewer_2nSf · 2024-11-04

**Soundness:** 2
**Presentation:** 2
**Contribution:** 2
**Rating:** 3
**Confidence:** 3

**Summary:**

This paper revists the idea of "entropy calibration" ([Braverman et al., 2020]) and proposes a new algorithm to calibrate the entropy of a language model:
- This paper examines the entropy of large language models (Llama 2 up to 70B, with and without instruction tuning), and shows that these models still exhibit a growing entropy rate, unlike human text's entropy rate which is lower and decreasing.
- This paper examines commonly used sampling methods (temperature, epsilon sampling, top-p sampling, top-k sampling), and shows that while these methods can achieve better entropy calibration with properly chosen parameters, the improvement comes at the cost of worse log-loss.
- This paper proposes a calibration algorithm that does not sacrifice log loss. The proposed algorithm is more practical than Algorithm 1 of [Braverman et al., 2020] and more accurate than Algorithm 2 of [Braverman et al., 2020].

[Braverman et al., 2020]: https://proceedings.mlr.press/v119/braverman20a.html "Calibration, Entropy Rates, and Memory in Language Models"

**Strengths:**

- (clarity, quality) The examination of entropy rate of Llama 2 at various sizes and with various sampling methods clearly demonstrates that growing entropy rate is still a relevant problem for LLMs, making this paper very well motivated.
- (originality, quality, significance) The proposed calibration algorithm is more practical than Algorithm 1 of [Braverman et al., 2020] and more accurate than Algorithm 2 of [Braverman et al., 2020].


[Braverman et al., 2020]: https://proceedings.mlr.press/v119/braverman20a.html "Calibration, Entropy Rates, and Memory in Language Models"

**Weaknesses:**

### Evaluation metrics

This paper presents extensive results when examining the issue of growing entropy rates with baseline models and sampling methods. It would be very useful if the proposed algorithm were also evaluated in the same manner. Currently, in Section 5.2, there is only an indirect evaluation of future entropy rates to justify the proposed method, and a small number of generation samples (from an uncalibrated model) are provided in the appendix with subjective qualitative evaluation. Providing a comparison in entropy rate in the same manner as Figures 1 & 4, as well as log loss will better support the claims in this paper.

### Novelty

The proposed algorithm can be viewed as an extension of Algorithm 2 of [Braverman et al., 2020]. The connection is not clearly discussed in the paper, and drawing the connection will help the readers better understand the contribution of this paper in context.

- In Algorithm 2 of [Braverman et al., 2020], the future entropy of only the immediate next step of the input model $\hat{p}$ is used in calibration and a single $\alpha$ is used to adjust weights at all time steps.
- In the proposed algorithm, the future context window is extended to $T$, the LLM's max sequence length, and a set of per-position $\{ \alpha_t \}$ is used instead.
- The proposed algorithm initially uses a recursive definition (Eq 9), but to improve practicality, an approximate surrogate
  $\hat{p}'_\alpha$

  (line 429) is instead used for the main results. This paper does not discuss in the choice of $\hat{p}'_\alpha$ in detail. It appears to me that the most obvious choice of that surrogate would be $\hat{p}$ just like Algorithm 2 of [Braverman et al., 2020].

### Experiment design

Algorithm 2 of [Braverman et al., 2020], while with a limited one-step lookahead entropy, is still an algorithm that should improve entropy calibration without hurting log loss. Therefore, the contribution of the proposed method would be much better quantatively understood if it could be compared with Algorithm 2 of [Braverman et al., 2020], especially given the close connection discussed above.

One of the main difference of the proposed algorithm from Algorithm 2 of [Braverman et al., 2020] is the future window size (1 vs $T$). It would be useful to also consider the possible window sizes in between for experiments, because smaller windows are cheaper computationally, and there might be a diminishing return in larger window sizes. Knowing the quality-compute tradeoff would be of great help to practioners.

The choice of $\hat{p}'_{\alpha}$ should also be evaluated in experiments. (Sorry, I had to break my paragraph here, otherwise the LaTeX markdowns mess up the text.)

While $\hat{p}$ appears to the most immediate choice, one can imagine an iterative process where the calibrated model from the previous iteration could be used as $\hat{p}'_\alpha$ for a new iteration to obtain a more accurately calibrated model. Perhaps that could be another way to get calibrated models more cheaply?

[Braverman et al., 2020]: https://proceedings.mlr.press/v119/braverman20a.html "Calibration, Entropy Rates, and Memory in Language Models"

**Questions:**

- A few models achieve lower entropy than the log loss of human text (e.g. Llama-2-70b-chat-hf in Figure 1; many params in Figure 4). However, the log loss of human text is only an upperbound of the entropy of human text, one cannot thus conclude that such a model is not entropy calibrated merely based on the information presented so far in the paper (admittedly these models are unlikely to be entropy calibrated given how big the gap there is between them and the log loss of human text; but we don't know the difference between the log loss and the true entropy of human text either). Would there be a method to better substantiate the claim of such models not being entropy calibrated?

- In a few places (lines 17, 44, 246), entropy is described with the phrase "i.e. confidence". Should it be "i.e. uncertainty" instead, since higher entropy means less confidence (more uncertainty)?

---

### Official Review · Reviewer_jcUt · 2024-11-04

**Soundness:** 2
**Presentation:** 2
**Contribution:** 2
**Rating:** 3
**Confidence:** 3

**Summary:**

The paper investigates the entropy calibration of language models. I believe the notion of entropy rate calibration, which the authors seek to study, is novel. Specifically, it states a language model is entropy rate calibrated if the entropy of the LM matches the entropy of the text. This condition seems to be equivalent to matching the true distribution under the continuous mapping theorem. Specifically, if I have a distribution p that gives rise to my training corpus and a language model p, if p = q, then H(p, q) = H(p) = H(q). Thus, I think the role of entropy rate calibration is one of a training diagnostic. The key finding asserts that current language models are miscalibrated.

**Strengths:**

The paper is certainly on an interesting topic. I was very intrigued when I was assigned it to review.

**Weaknesses:**

Parts of the notation are very precise. The first equation in the paper (I wish it were labeled so I could reference it) is hard to understand. It's unclear whether the capital letters are strings or random variables. When the authors write Y ~ p(Y | X), I don't know what that means really. I think it means that Y is a random variable distributed according to p(Y | X), but I am not sure whether the prompt is fixed or not, i.e., do they mean Y ~ p(Y | X = x)? I think it's clarified later in the paper that X is a fixed prompt, i.e., a string, but then the notation seems very inconsistent.

The vocabulary cannot include EOS, in general, so the function phat should be of size |V| + 1 in section 3.

It was very hard to follow much of Section 3. I suspect this is core to understand the method at a deeper level.

**Questions:**

I am actually confused about how the authors estimate entropy. Is next-symbol entropy used as a project? The entropy discussed in the introduction involves a sum over V* that cannot be done without sampling. If the entropy is approximated, I would have expected to see a number of plots vetting the approximation.

---

### Note · Authors · 2024-11-15

**Comment:**

Thanks to the reviewers for the valuable feedback. We will use this feedback to design a more complete set of experiments to improve the paper.

**Withdrawal Confirmation:**

I have read and agree with the venue's withdrawal policy on behalf of myself and my co-authors.